# Mucosal Melanoma: Mechanisms of Its Etiology, Progression, Resistance and Therapy

**DOI:** 10.3390/cells14231884

**Published:** 2025-11-27

**Authors:** Sofie-Yasmin Hassan, Thomas W. Flanagan, Sarah-Lilly Hassan, Sybille Facca, Youssef Haikel, Mohamed Hassan

**Affiliations:** 1Pharmacy, Heinrich Heine University Duesseldorf, Universitaetsstr. 1, 40225 Duesseldorf, Germany; sofie00@gmx.de; 2Department of Pharmacology and Experimental Therapeutics, LSU Health Sciences Center, New Orleans, LA 70112, USA; tflan1@lsuhsc.edu; 3Department of Pharmacy, University of Bonn, An der Immenburg 4, 53121 Bonn, Germany; slh03122001@gmail.com; 4Icube Laboratory, CNRS, UMR 7357, University of Strasbourg, 67000 Strasbourg, France; sybille.facca@unistra.fr; 5Institut National de la Santé et de la Recherche Médicale, University of Strasbourg, 67000 Strasbourg, France; youssef.haikel@unistra.fr; 6Department of Operative Dentistry and Endodontics, Dental Faculty, University of Strasbourg, 67000 Strasbourg, France; 7Pôle de Médecine et Chirurgie Bucco-Dentaire, Hôpital Civil, Hôpitaux Universitaire de Strasbourg, 67000 Strasbourg, France; 8Research Laboratory of Surgery-Oncology, Department of Surgery, Tulane University School of Medicine, New Orleans, LA 70112, USA

**Keywords:** mucosal melanoma, melanocytes, check point inhibitors, immune therapy, c-Kit

## Abstract

**Highlights:**

**What are the main findings?**
Both cutaneous melanoma (CM) and mucosal melanomas (MM) differ significantly in their epidemiology, genetic profile, and clinical presentation, while their treatment options are similar.The response rates to treatment and consequently the survival rate are significantly lower in MM than in CM.

**What are the implications of the main findings?**
MM exhibits unique clinical and molecular characteristics that can lead to resistance mechanisms not observed in CM.Compared to available anticancer agents, immunotherapy is the best treatment option for MM.

**Abstract:**

Mucosal melanoma (MM) is a rare, aggressive cancer whose incidence has increased continuously over the years. This subtype of melanoma arises from melanocytes on hairless surfaces, typically in the respiratory tract, gastrointestinal (GI) tract, and urogenital tract. The most common sites of occurrence include the head and neck, the anorectal region, and the vulvovaginal region, while the rare sites of MM are the urinary tract and the upper and lower GI tract, including the esophagus, duodenum and the gallbladder. MM arises in melanocytes of the ectodermal mucosa that originate from neural crest cells and migrate through embryonic mesenchyme to their destination. Although melanocytes are located mainly in the epidermis and dermis, their presence in various extracutaneous sites, such as the eyes, mucosal tissue, and leptomeninges, is known. Although both cutaneous melanoma (CM) and MM differ in their epidemiology, genetic profile, and clinical presentation, their treatment options are similar. In contrast to the higher treatment response of CM, MM is characterized by a lower response rate to available treatment options, resulting in a poorer survival rate. In this review, we provide an overview of the biology of MM and the mechanisms regulating its development, progression and treatment resistance.

## 1. Introduction

Mucosal melanoma (MM) is a rare, aggressive cancer whose incidence has steadily increased over the years [1,2]. MM arises from melanocytes on hairless surfaces, typically in the respiratory tract, gastrointestinal (GI) tract, and urogenital tract [2,3]. The most common sites of MM occurrence are the head and neck (55.4%), the anorectal region (23.8%), and the vulvovaginal region (18.0%) [4,5], while the rare sites of its occurrence are the urinary tract and the upper and lower GI tract, including the esophagus, duodenum and the gallbladder [5,6].

Melanocytes are mainly located in the epidermis and dermis, but they also occur in various extracutaneous sites, such as the eyes, mucosal tissue, and leptomeninges [3,7] and its development occurs in the melanocytes of the ectodermal mucosa [2,8]. Although the treatment options of both cutaneous (CM) and MM are similar, their epidemiology, genetic profile, clinical presentation, and treatment response are different [1,2].

In contrast to CM, MM is characterized by a lower response rate to available therapeutics, resulting in a worse survival rate [9,10].

Although the main function of melanocytes in the mucosal membranes is not well addressed, their antimicrobial and immunologic functions have been proposed [11,12,13].

The pathogenesis of MM is not described in detail. The oncogenic mutations of the BRAF gene that are characteristic of CM are not considered to be crucial for the development of MM [14,15]. In contrast, mutations or increased copy number of KIT have been described in several studies as a common phenomenon in MM [16,17,18].

As one of the most aggressive subtypes of melanoma, MM has a worse prognosis than those of the CM [18,19]. Data from a retrospective study showed that [20,21]. Although the mechanisms of the poor prognosis of MM are unknown, the possible reasons for this phenomenon lie in the biological pattern of mucosal melanocytes as well as in the risk factors associated with MM incidence [22,23,24,25]. In addition to the crucial role played by chronic inflammatory diseases, viral infections, and chemical irritants in the development of vulvar MM, infection with the human immunodeficiency virus (HIV) is also crucial for the development of anorectal MM [23,26,27].

Unlike the development of CM, the evolution of MM from precursor lesions is not reported [13,28]. It is noteworthy that the development of CM is usually associated with various types of precursor lesions, including benign melanocytic nevi, which are usually characterized by the common BRAF V600E mutation, and dysplastic nevi, which are characterized by significant NRAS alterations and mutations in the TERT promoter [29,30]. The transformation of benign lesions into CM is due to the loss of CDKN2A, PTEN and TP53 functions via a mutation-dependent mechanism, leading to enhanced tumor progression, invasion and metastasis [31,32]. Although several forms of benign melanocytic mucosal lesions have been reported, the lack of significant criteria to identify MM precursor lesions is the reason why the understanding of the evolutionary course of MM is limited despite the molecular profiles revealed by whole-genome sequencing [1,22].

In this review, we summarize the current findings on the genetic and pathological development of MM from benign melanocytic mucosal lesions to malignant tumors. In addition, the role of functional mutations and signaling pathways in MM progression and treatment resistance as well as possible therapeutic options for MM will be highlighted. Furthermore, this review focuses on the mechanisms of resistance to ICIs, the more effective treatment for MM.

## 2. Epidemiology

The epidemiology of MM is quite different from those of its cutaneous counterpart, indicating that the biology of both MM and CM is significantly different [2,33]. While the development of MM is common in elderly individuals, with a median age of 70 years by the first diagnosis, when compared to CM that occurs in younger individuals with a median age of 55 years [34,35].

Although the incidence of CM is continuously in an increase, when compared to the incidence rate of any other cancer types in the United States, West Europe and Australia, The stable incidence of MM is not related to the lower incidence of the disease, but rather to the limitation of available diagnostic techniques for identifying mucosal melanomas in hidden areas of the body [34,36]. Of note, MM is usually only diagnosed when the disease is already well advanced, and treatment outcomes are poor [1,2].

The main limitations affecting mucosal melanoma staging are attributed to the lack of a standardized, and specific system for mucosal melanoma staging [17,37]. Other factors that complicate staging include the use of systems for other cancers, which may not be suitable for staging mucosal melanoma, the specific anatomical location of the tumor, its aggressive behavior, and the tendency for late diagnosis [32,37]. Although there are ongoing efforts to develop specific risk-based tiered models, the lack of a generally accepted system represents a major limitation [17,38].

The 5-year overall survival rate for MM is approximately 25%, compared to the 5-year overall survival rate for CM, which is estimated to be approximately 80% [39,40]. The worse the treatment outcomes of MM, the more advanced the disease is at the time of diagnosis [2,41]. The anatomical features of MM also represent an obstacle to its complete resection, apart from the fact that the mucosal surfaces are well supplied with lymphatic vessels [42]. In addition to the lower survival rate and based on the development of clinical metastases, patients with MM also have a poor stage compared to other melanoma subtypes [3,41].

While CMs occur more frequently in men than in women, the incidence rate of MM is higher in women than in men, and this observation is presumably due to the higher incidence rate of genital tract melanoma in women [2,3,28]. Ethnic differences in the incidence of MM have been reported in several studies, with the highest incidence rate of MM being in Chinese patients with 23% of melanomas [33,43], followed by 5 to 13% of melanomas in Black patients [2,3,44], and the lowest incidence rate of MM being in White patients with approximately 1 to 2% of melanomas [37,45]. While the occurrence of CM is attributed to exposure to ultraviolet (UV) radiation, the occurrence of MM is related to hereditary rather than environmental risk factors [36].

## 3. Common Sites for the Occurrence of Mucosal Melanomas

As a rare type of melanoma, MMs occur most frequently in the head and neck region, the anorectal area, and the female genital tract. In particular, the nasal cavities and paranasal sinuses, the oral cavity and the anorectal regions [46,47]. The region of head and neck is the most common site for MM in men than in women, while the incidence of genitourinary melanomas is higher in women than in men [1,2,48]. Accordingly, the genitourinary melanomas account for more of 50% of MMs and up to 7% of all melanomas in women [25,48].

More than 70% of the head and neck areas are affected by sinonasal melanomas, with 80% occurring in the nasal cavity [21,49]. The areas of the sinuses most affected by MM are the maxillary sinus, as well as the ethmoid, frontal, and sphenoid sinuses [50]. While 70% of head and neck MM are due to sinonasal melanomas, the other 30% of head and neck MM are due to melanomas of the oral cavity [51,52]. The most melanomas of oral cavities occur mainly in the palate and maxillary gingiva [53,54]. The anatomical structure of the mucous membrane of the head and neck consists of a layer of stratified squamous epithelium and an underlying layer of connective tissue, the so-called lamina propria [55]. The regulation of oral mucosal immunity is mediated by local and regional lymph nodes, such as those of the tonsils, adenoids, and cervical lymph nodes, and not by the lymphatic tissue associated with the mucosa [56]. The mucosal surfaces of the oral cavity contain Langerhans cells, which are characterized by higher expression of Toll-like receptors (TLR)2 and TLR4 on their surfaces [22,26]. Through the activation of TLRs by the action of lipopolysaccharides (LPS), oral Langerhans cells can upregulate the expression of the coinhibitory molecules B7-H1 and B7-H3 and downregulate the expression of the costimulatory molecule CD86 (B7-2) [57]. As a result, regulatory T cells can develop and subsequently secrete interleukin (IL)-10 and transform growth factor beta (TGF-ß) to suppress the immune response [58].

Vulvovaginal MM that belongs to the genitourinary cancer and accounts to the second most common cancer of the vulva after squamous cell carcinoma in women [59]. While more than 80% of vulvar melanomas arise in the labia minora, the clitoris, or the inner, glabrous, hairless part of the labia majora, the incidence of vulvar melanomas in the outer, non-glabrous, hairy part of the labia majora is less than 20% [60,61]. It is known that the mucosa of the female genital tract differs considerably from that of the gastrointestinal tract in terms of its cellular composition, microflora, and innate and adaptive immunity-dependent mechanisms [60,61]. Of note, the biology of the genital, anal, and lower rectal tract is not described in detail compared to that of the rest of the gastrointestinal tract. As reported, the mucous membranes lining the surfaces of the female genital tract can be divided into two different types separated by a line at the transformation zone [44,62]. While type I mucosal surfaces consist of a single layer of densely connected, simple columnar epithelium lining the uterus, cervix, and upper rectum [63,64], type II mucosal surfaces consist of multiple layers of stratified squamous epithelium lining the external and internal vagina and the ectocervix [63,64]. As a stratified squamous epithelium, type I is functionally responsible for replacing the missing tight junction and thus enabling the free transport of molecules [14,63].

While anorectal melanoma accounts for about 0.2 to 1.3% of all melanomas and about 1% of all anorectal carcinomas [65,66]. While about 39 to 42% of anorectal melanomas are in the rectum, about 30% arise in the anal canal and the site of origin of the rest is unknown [65,66].

In contrast to other mucosal melanomas, the incidence of anorectal melanomas has increased over the years [67,68]. Like the mucous surfaces of the vulvovaginal region, the mucous surfaces of the anorectal canal are organized into type I and type II tissues [69,70].

While type I mucosal surfaces line the upper rectum and consist of a single layer of densely connected, simple columnar epithelium, type II mucosal surfaces line the lower rectum and anus and consist of several layers of stratified squamous epithelium [69,70]. In contrast to the type I cervix, whose mucus comes from glands in the crypts, the type I rectum is lined with goblet cells. In addition, type II anorectal surfaces are drained by the inguinal lymph nodes due to the lack of mucosa-associated lymphatic tissue [69,70]. The structure of Type I and Type II is outlined in Figure 1.

## 4. Biology of Melanocytes and Their Function

Melanocytes are neural crest-derived cells that migrate to skin, eyes, leptomeninges, and mucous membranes during the development [6]. Skin melanocytes have two final destinations, including the hair follicles and the basal cell layer of the epithelium, where the migrated melanocytes fulfill their main biological function of melanin production [6,11]. The main function of melanin produced by melanocytes is to protect the skin from UV radiation and to scavenge cytotoxic free radicals produced by sun exposure [71,72]. Exposure to the sun causes melanin secretion to occur near the keratinocytes, thereby protecting the keratinocyte genome from sun damage [73,74].

Melanocytes are residing not only in the skin but also in many sun-protected mucous membranes, including the respiratory tract (mouth, nose, throat, larynx and upper esophagus), the intestine, the urogenital tract and the rectal tract [1,75]. Since melanocytes in mucous membranes are not normally directly exposed to sunlight, it is unlikely that photoprotection is the primary and final function of mucosal melanocytes [3,22]. Some reports have shown that melanocytes located in mucosal tissues are due to errors in migration from the neural crest during embryogenesis [11]. However, there is increasing evidence that mucosal melanocytes may have biological functions in addition to pigment production [41,76]. Since the mucosa plays an important role in the innate immune defense system, mucosal melanocytes are also thought to have immunogenic functions [1,3]. In addition to their function of protecting the skin from UV radiation and its biological consequences, melanocytes are characterized by their ability to bind toxins and thereby neutralize toxins produced by bacteria [77,78]. Melanocytes are known for their antimicrobial activity leading to the disruption of the lipid bilayer of cell membranes of bacterial pathogens [78,79]. Remarkably, the antimicrobial activity of melanocytes is attributed to the production of aromatic precursors, including quinone and semiquinone intermediates, which are formed during the myelinization cascade [72,79].

Since mucosal melanocytes are largely protected from light, it is expected that they produce a limited amount of melanin [11]. Consequently, mucosal melanocytes are unable to exert sufficient antimicrobial activity in their microenvironment.

In addition to the antibacterial properties of melanin, melanocytes are also involved in the regulation of the intrinsic and acquired immune system [80]. The participation of melanocytes in innate immunity is mediated by Toll-like receptors (TLRs) that they express [80,81]. The elimination of microbial pathogens such as bacteria and fungi by melanocytes occurs via a phagocytosis-dependent mechanism [82,83]. Melanocytes engulf microbial pathogens by recognizing microbially associated molecular patterns in a TLR-dependent manner [84,85]. Melanocytes can serve also as a component of acquired immunity [86,87]. The expression of major histocompatibility complex (MHC) class II by melanocytes provides evidence for their ability to function as antigen-presenting cells (APCs) [88,89].

In contrast to cutaneous melanoma, which can develop from various benign and pathological stages such as benign nevi, dysplastic nevi and malignant tumors in situ [90,91], the mechanisms of MM development are still unknown. To date, there is no clear definition and characterization of the precursor lesions of MM that could be reported [1,22]. Melanocytic nevus, which is the medical term for a mole and can appear anywhere on the body as a non-cancerous skin lesion and does not require any treatment [29]. Therefore, the development of melanocytic nevi, including macular, nevus and melanoacanthomas into melanoma is rare [22,29]. As one of the most common melanocytic lesions, mostly known as lentigo simplex, which appears in the form of a group of small and round macules [84,92]. Macules are characterized by a variation in their color range from gray to brown to black [93,94]. However, the diversity of pigmentation of macules is determined by the ratio of eumelanin and pheomelanin [95,96,97,98]. As benign lesions, macules are characterized by melanin deposition and the loss of Ki-67 [97,99]. Based on histological studies, the basal cell layer of benign macules is characterized by its uniform melanin accumulation, regardless of the density of melanocytes or the presence of nevus [100]. Apart from their asymptomatic nature, macular lesions show no evidence of malignant transformation [101,102]. The most common molecular lesions in the oral cavity include the vermillion border of the lips, the gums and the alveolar ridge, as well as the buccal or lip mucosa [93,103]. Although no direct association between melanotic macules and the development of oral mucosal melanoma (OMM) has been reported, several case reports have described the transformation of benign macules into malignant OMM. This provides evidence of the malignant potential of some macular lesions as precursors to the development of OMM [22,101]. While the oral nevi are much less than the nevi on the skin, the transformation of oral nevi into OMM is considered. As is known, oral nevi are mainly divided into three types including, the junctional nevi, which are in the extended epithelial tips; the compound nevi, which are organized in nests and belts in the lamina propria; and the subepithelial nevi [104,105]. The development of nevi in the oral mucosa results from the proliferation of melanocytes together with the epithelial basal cell layer [93,105]. Like macules, oral nevi are characterized by clear margins and usually have raised pigmented lesions [93,106]. Therefore, the presence of dysplastic nevi is expected to be a risk factor for the development of OMM [90,107]. In contrast to normal nevi with macular or popular components and undetermined borders, dysplastic nevi are characterized by their large diameter [89,91]. Compared to macule and nevus lesions, melanoacanthoma is a rare benign melanotic lesion that can mimic OMM based on its rapid development [108]. In OMM patients, benign pigment disorders are initially diagnosed before benign lesions develop into malignant disease [109,110].

Preferentially expressed Antigen in Melanoma (PRAME), a multifunctional cancer testicular antigen expressed in normal and neoplastic tissue, has proven to be a useful diagnostic tool in the differential diagnosis between benign and malignant melanocytic lesions [111,112]. In addition to be a promising target for immunotherapy, PRAME was approved as a diagnostic marker for subgroups of mucosal melanoma, including urological, gynecological, and head and neck mucosal melanomas, due to its reliability [113,114]. The differential expression of PRAME in different subtypes of mucosal melanoma therefore makes it useful to distinguish malignant melanomas from their benign counterparts [112,115].

## 5. Mechanisms of Mucosal Melanoma Development and Progression

Despite the common origin of CM and MM, the clinical and pathological manifestations of MM differ significantly from those of CM [2,22]. Ultraviolet exposure is unlikely to contribute to the development of MM [22,28,116]. Because the localization and nonspecific symptoms of MM prevent early detection, MMs are usually diagnosed late, especially when most cases are already in an advanced stage [2,3]. Therefore, the etiopathogenesis, identity of precursor lesions and biology of MMs are not fully understood [11,117]. Consequently, mucosal melanoma has a worse prognosis and effective treatment options are lacking [11,22,118]. The current treatment of MM is usually the same as that of CM, the validation of the therapeutic approaches of CM for MM treatment is still lacking [117,118,119]. Despite the technological developments in surgery and radiotherapy as well as advances in systemic modalities, no increased survival benefit has been observed in MM [120,121]. Also, the heterogeneity of MMs that is related to their anatomical location is a further challenge for the management [46,122].

The development and progression of MM are regulated by a complex process mediated by various mechanisms, including genetic and epigenetic alterations that lead to aberrant regulation of signaling pathways and significant alteration of the tumor microenvironment [123,124].

In contrast to CM, the development of MM is not driven by UV light, but rather is the consequence of the abnormal activation of important signaling pathways such as MAPK, PI3K/AKT, and WNT as well as to active mutations that frequently occur in KIT, NF1, and BRAF genes [123,124].

While CM is genetically controlled by BRAF mutants (50%), RAS mutants (20–30%), NF1 mutants (10–15%) or triple wild types (10–15%) [123,124], the MM is driven by SF3B1-mutant (15%), NF1 mutant (14%), KIT-mutant (13%), NRAS-mutant (8%) and BRAF (6%)-dependent mechanisms [13,124].

Although MMs may share some common characteristics, as they all fall under the broader category of mucosal melanomas [1,2], MMs of different sites, including those in the head and neck area as well as those of urological and gynecological mucosal origin, are molecularly different [1,2]. They differ in their specific mutation patterns, such as the higher frequency of TP53 mutations and the absence of BRAF mutations in urethral melanoma compared to CM [125,126]. They also differ in other genetic alterations such as KIT and NRAS mutations, which are known to affect MMs response to targeted therapies [10,124]. For example, urethral melanomas exhibit a high frequency of TP53 mutations and a remarkable lack of BRAF mutations [125,126]. Although TP53 and KIT mutations are observed in gynecological melanomas, BRAF and NRAS mutations have been shown to occur in some cases [127,128].

Mutations in the RAS/BRAF axis system, especially NRAS, are common in sinonasal melanomas, often even more common than other common melanoma mutations such as BRAF [123,129]. Some studies suggest that RAS mutations, particularly in the NRAS gene, are the most common driver mutations in sinonasal melanomas [123,124]. Although BRAF is the most frequently mutational oncogene in cutaneous melanoma, studies of sinonasal melanomas show a higher prevalence of NRAS mutations compared to BRAF mutations [130,131].

The progression of MM including, metastasis, and immune evasion is mediated by molecular mechanisms, which can be clearly distinguished from those of CM [41,132]. In contrast to CMs, MMs are characterized by a lower incidence of somatic mutations, as neither UV-induced mutational signatures nor oncogenic fusion genes are present [13,133]. For example, signal transduction via the mitogen-activated protein kinase (MAPK) pathway is involved in the activation of the proteins RAS, RAF, MEK, and ERK [130,134]. As widely documented, mutations in the key molecules BRAF, NRAS or NF1 are characteristic of most CM (94%) and are reported in only about 28% of MM [130,134].

KIT mutations are significantly more common in MM than in CMs, with a rate of 13–39% in all MMs [135,136]. The common mutations/amplifications of the KIT gene were identified in anorectal and urogenital melanomas [137,138]. Furthermore, KIT mutations are mutually exclusive with NRAS, BRAF, and NF1 mutations and are involved in the activation of the MAPK signaling pathway through different pathways [127,139].

While the mutations in the GNAQ gene, which encodes the guanine nucleotide-binding protein G(q) subunit α, account for 1–12% of MM cases, the mutation in GNA11 accounts only for 1% [140,141].

The role of the PTEN/PI3K/AKT/mTOR signaling pathway in both CM and MM has been discussed in several studies [2,13,41,142]. While the mutation or loss of PTEN (phosphatase and tensin homolog), which is observed in up to 8.5–40% of CM patients, has also been reported in 1–25% of MM patients [143,144]. Therefore, patients with TP53 mutations and PTEN alterations also had a poor survival rate in MM [1,14,117].

Rapamycin-insensitive companion of mTOR (RICTOR) regulates signaling pathways such as Akt via mTORC2 and thus influences cell growth and proliferation. However, the most common genetic alteration in oral mucosal melanoma (OMM) is associated with RICTOR [122,145].

As reported in several studies, the abnormal copy-number variants in CDK pathway genes are most common in MM [122,146]. While the observed amplification rates for CDK4 and CCND1 in MM are 47.0% and 27.7%, respectively, an increased deletion rate of 57.7% was also reported for P16INK4a [146,147]. However, better outcomes have been reported in MM patients with CCND1 amplification [117,148]. The potential efficacy of CDK4 inhibitors in MM is also promising [117,147]. In addition to the increased deletion rates of CDKN2A and CDKN2B in MM [117,145], the amplification of MYC, kDR, KIT, PDGFRA, MITF, VEGF and CCND3 is also common [14,117].

The role of spliceosomes pathway in the pathogenesis of MM is considered. Since mutations in spliceosome factor 3b (SF3B1), which encodes the largest subunit of the SF3B protein complex, have been shown to be involved in the pathogenesis of many cancers, including MM [149,150]. In addition, genome sequencing data identified SF3B1 driver mutations as predominant in MM [149,150]. Moreover, higher mutation frequency of SF3B1was observed in MM [149,150]. However, all detected mutations have been reported to occur at a hotspot in codon 625, mainly at anorectal or female genital sites [149,150]. While the SF3B1 R625 codon mutations seem to be unique to MMs [149,150].

The main genes and their functional roles in the development of MM progression is outlined in Figure 2.

## 6. Therapeutic Option of Mucosal Melanoma

MM exhibits heterogeneity on several levels, including genetic, molecular, and phenotypic differences between different tumors as well as within a single tumor [1,113]. This is reflected in the diverse genetic mutations, which differ from those of cutaneous CM and vary depending on the tissue of origin, as well as in the different cellular subpopulations with varying metastatic potential and metabolic profiles [151,152]. Consequently, the frequent occurrence of heterogeneity between different subgroups of mucosal melanoma, as well as within a single tumor, complicates treatment and underlines the need for personalized strategies based on individual tumor analyses [40,153].

Although numerous technological developments in surgery, radiotherapy, and systemic treatment modalities, no increased survival benefit was observed in MM [150,151,154,155]. The heterogeneity of MM at different sites presents a challenge for treatment.

However, MM poses a significant challenge in achieving negative resection margins due to its advanced stage at diagnosis, frequent lymph node involvement, complex anatomical locations and multifocal lesions [122,156]. Consequently, surgical treatment of MM is usually associated with a recurrence rate of 50% to 90% [50,117,148]. Thus, the treatment of MM requires a multidisciplinary approach particularly for complex anatomical locations and multifocal lesions [26,157]. However, neoadjuvant and adjuvant therapies are expected to improve outcomes in resectable MM [158,159]. While the most common treatment options for advanced or metastatic disease are radiotherapy, immunotherapy and targeted therapy [41,160]. The therapeutic management of anorectal melanoma is outlined in Figure 3.

### 6.1. Immune Therapy

Although the first attempts to use the immune system to fight cancer were only made towards the end of the 19th century, significant progress has been made first in recent years [161]. However, the elucidation of the immune mechanisms of multiple myeloma paved the way for the accelerated development of innovative immunotherapeutic strategies. Immune checkpoint inhibitors (ICIs) have proven to be promising agents capable of disrupting immune checkpoint interactions, thereby effectively activating the host’s immune system leading to the destruction of tumor cells [148,149]. In parallel with ICIs, other forms of therapy such as adoptive cell therapy (ACT), vascular endothelial growth factor (VEGF) inhibitors and combination therapies have also shown a strong anti-tumor effect [150,154]. Clinical trials investigating the efficacy of ICIs include anti-CTLA-4 and anti-PD-1 therapies alone or in combination.

While immunotherapy, especially with PD-1 inhibitors, is the more effective therapeutic approach for metastatic and surgically unresectable MM [162,163]. Even though the treatment responses are lower in MM than in CM [9,10,41].

Clinical studies of VEGF inhibitors in MM predominantly involve combinations with PD-1 inhibitors and show promising results such as objective response rates (ORR) of about 40–45% in early studies [122,157].

PD-1 inhibitors form a pharmacological subclass of therapeutics that specifically target the immune checkpoint protein PD-1 and thereby inhibit the interaction between PD-1 and its ligands [157,158]. These remarkable advances in immunotherapy herald a new era in cancer treatment, including the treatment of MM [104,155]. This inhibition restores and enhances the immune response of activated T cells against tumor cells [160,162]. Importantly, PD-1 inhibitors have demonstrated considerable efficacy in patients with MM, as evidenced by several clinical trials that confirm their remarkable performance in MM therapy [163,164]. These studies highlight the favorable and sustained results of PD-1 inhibitor monotherapy and offer survival benefits for patients with advanced and metastatic melanoma [165,166]. In addition, a study on the efficacy of anti-PD-1 therapy in patients with head and neck mucosal melanoma (HNMM) was conducted, in which the expression of PD-L1 and PD-1 in tumor tissue samples was examined [167,168]. Although a low PD-L1 expression level was observed in HHMM samples, treatment with anti-PD-1 therapy did not lead to clinical success in these patients [147,155]. This is evidence that highlights the challenges and limitations of checkpoint inhibitors in the treatment of advanced MM.

As is reported, CTLA-4 inhibitors mediate their effect mainly through the specific inhibition of the CTLA-4 protein expressed on the cell surface, the essential modulator of T-cell function [169,170]. These inhibitors mediate their antitumor activity by inhibiting the interaction between CTLA-4 and its ligands (B7-1 and B7-2), which in turn suppresses inhibitory signals to T cells and leads to enhanced antitumor immune responses [136,171]. Although initial clinical studies have consistently shown remarkable therapeutic efficacy of CTLA-4 inhibitors in patients with MM, Patients with advanced mucosal melanoma who had failed nivolumab therapy [104,155]. In contrast, treatment with ipilimumab as an adjuvant therapy has been reported to offer additional benefits in terms of progression-free survival in patients with advanced, nivolumab-resistant mucosal melanoma [155,172,173,174,175].

The PD-1/PD-L1 and CTLA-4 signaling pathways are important immune checkpoint pathways with different mechanisms of action in cancer therapy [171,176]. Therefore, the combination of PD-1/PD-L1 and CTLA-4 inhibitors may be more effective than monotherapy in MM [136,177]. Although increasing evidence suggests that combining ICIs enhances the immune response and improves survival, a frequent increase in the risk of serious immune-related adverse events has been observed [178,179]. The combination therapy, which often includes drugs such as nivolumab (anti-PD-1) and ipilimumab (anti-CTLA-4), works by simultaneously ‘priming’ and ‘enhancing’ T-cell responses to release an antitumor potential against the tumor [33,180]. Although mucosal melanomas generally have lower response rates than other melanoma subtypes, combination therapy is an important area for future research to overcome resistance and control toxicity.

Of note, the highest 5-year OS rate among therapies for advanced melanoma was observed with combined anti-CTLA-4 and anti-PD-1 checkpoint inhibition, which is most strongly associated with the highest incidence of immune-related adverse events [176,181]. Data from comparative clinical trials of monotherapy and combination therapy with nivolumab and ipilimumab showed that combination therapy offers advantages.

Clinical studies of VEGF inhibitors in MM predominantly involve combinations with PD-1 inhibitors and show promising results such as objective response rates (ORR) of about 40–45% in early studies [123,182]. These studies use drugs such as axitinib (a VEGFR inhibitor) and nivolumab or toripalimab (PD-1 inhibitors). It has been reported that combining anti-VEGF therapy with PD-1 blockade could be more effective than single-drug immunotherapy or chemotherapy alone [182,183]. Some studies have also examined chemotherapy in combination with bevacizumab (a VEGF inhibitor) [182,184].

The highest 5-year OS rate among therapies for advanced melanoma was observed with combined anti-CTLA-4 and anti-PD-1 checkpoint inhibition, which is most strongly associated with the highest incidence of immune-related adverse events [176,181]. Data from comparative clinical trials of monotherapy and combination therapy with nivolumab and ipilimumab showed that combination therapy offers advantages over monotherapy, as demonstrated by the significant increase in the median PFS and objective response rates (ORRs) [185,186].

Although the median response time was similar in MM and CM, apart from the type of treatment, patients receiving nivolumab monotherapy showed lower median PFS of 3.0 months in MM and 6.2 months in CM, with ORRs of 23.3% and 40.9%, respectively [178,179]. In addition, patients who received ipilimumab monotherapy had a median PFS of 2.7 months for MM and 3.9 months for CM [178,179]. While patients receiving nivolumab in combination with ipilimumab had a better median PFS of 5.9 months for MM and 11.7 months for CM, with ORRs of 37.1% and 60.4%, respectively [180,187].

In summary, although the responses of MM to ICIs have been observed in several studies, the efficacy of ICIs in MM is relatively suboptimal compared to CM and other melanoma subtypes. However, ICIs remain the best therapeutic option for MM.

Due to its anatomical location, MM is characterized by a high tendency to vascular invasion, the mechanism that can relatively enhance its sensitivity to antiangiogenic agents [182,188,189]. Consequently, combination therapy based on antiangiogenesis could offer a therapeutic opportunity.

The combination of the toripalimab, a humanized anti–PD-1 monoclonal antibody, in combination with the inhibitor of vascular endothelial growth factor (VEGF) receptor, axitinib for advanced MM showed an ORR of 48.3% and a disease control rate (DCR) of 86.2% [182,190]. In addition, the median duration of response (DoR) was 13.7 months, the median PFS was 7.5 months, and the median OS was 20.7 months, as shown in a Phase 1b study [183,191].

While in phase II trial evaluation of atezolizumab, the anti-PD-1 monoclonal antibody with the VEGF monoclonal antibody, bevacizumab in advanced MM revealed an ORR of 36.4%, with a median PFS of 5.2 months and a DCR of 59.1% [46,192].

The evaluation of the efficacy and safety of the humanized anti-PD-1 antibody camrelizumab in combination with the VEGFR tyrosine kinase inhibitor apatinib in advanced MM showed an ORR of 42.9%, a DCR of 82.1%, and a median PFS of 8.05 months [193,194].

In addition, the combination of chemotherapy with antiangiogenic agents has been approved due to its efficacy in the treatment of unresectable or advanced MM [195,196].

### 6.2. Radiotherapy

Adjuvant radiotherapy is a postoperative treatment for mucosal melanoma that is often recommended to reduce the risk of local and regional recurrence, especially in patients with high-risk factors such as positive resection margins, advanced stage, or multiple positive lymph nodes [165,166]. Although adjuvant radiotherapy can improve local control, its impact on overall survival in MM remains controversial and should perhaps be considered in conjunction with modern systemic therapies [197,198].

Radiotherapy is often used as an adjuvant treatment in the treatment of MM, especially for postoperative lesion control or when surgery is not possible [197,198]. Although radiation therapy is effective in killing cancer cells, it can damage surrounding normal tissue [164,165]. Chemotherapy is a systemic treatment option for advanced or metastatic MM that aims to relieve symptoms and slow tumor growth [166,167].

It has been shown that radiotherapy promotes the infiltration of T cells into the tumor microenvironment and acts synergistically with the immune system by inducing the release of tumor-associated antigens [199,200]. This can lead to the local death of tumor cells and simultaneously trigger an abscopal effect, in which immune cells attack distant, non-irradiated tumor sites [201,202]. However, the efficacy of pembrolizumab in combination with hypofractionated radiotherapy in patients with mucosal melanoma is being investigated in the clinical trial [203,204]. Even untreated patients with primary sinonasal mucosal melanoma showed promising survival benefits from the combination of surgery, radiotherapy and immunotherapy, as demonstrated in a phase II trial of nivolumab and axitinib [205,206]. To that end, the combination therapy significantly improved progression free survival (PFS) and overall survival (OS) [207,208]. This was demonstrated by a Phase III clinical trial in patients with advanced BRAF V600E-mutated melanoma, based on the investigation of the safety and efficacy of combination therapy with BRAF and a MEK inhibitors [169,170].

### 6.3. Targeted Therapies

Targeted therapies, especially for specific mutations such as BRAF or KIT have been suggested for MM treatment [128,164].

In contrast to CM, MM has a higher KIT mutation rate, which may improve its response to KIT inhibitors compared to other melanomas [13,136]. There are no clinical trials specifically investigating KIT inhibitors in MM, most studies included a significant proportion of non-CM subtypes, with MM accounting for 46% to 71% of included cases [171,175,176,177]. Although imatinib represents a treatment option for patients with KIT mutations, its use is often limited to clinical trials [209,210]. In addition, many other targeted therapies are being investigated in clinical trials, sometimes in combination with other treatments such as immunotherapy [211,212]. The development of therapeutic approaches, including neoadjuvant and adjuvant therapies, is critically evaluated for the treatment of melanoma subtypes including MM [117,213].

The BRAF V600E mutation rate of MM is estimated at approximately 12% and could therefore benefit from combination therapy with BRAF and a MEK inhibitor [117,214,215].

The most frequently used targeted therapies for mucosal melanomas are BRAF and MEK inhibitors, which can be administered in combination, e.g., brafenib plus tramatinib, especially in patients with a BRAF V600 mutation [117,160]. However, if the melanoma has a C-KIT mutation, targeted therapies such as imatinib or nilotinib can be used [40,216].

In contrast to CM, MM has a higher KIT mutation rate, which may improve its response to KIT inhibitors compared to other melanomas [13,136,171]. Although there are no clinical trials specifically investigating KIT inhibitors in MM, most studies included a significant proportion of non-CM subtypes, with MM accounting for 46% to 71% of included cases [171,177].

The loss of phosphoinositide-dependent kinase-1 (PDPK1) showed clinical relevance for the therapeutic efficacy of MEKis in melanoma, as demonstrated by clinical studies [172,208]. However, the synergistic effect of PDPK1 depletion and MEKi was further investigated in melanoma cell lines with NRAS-mutation [172,173]. The clinical relevance of the combination of PDPK1 and MEK inhibitors was also demonstrated in the NRAS-mutated xenograft model, indicating the efficiency of the combinatorial strategy for the treatment of melanoma patients with NRAS mutation [172,174]. Although immune checkpoint inhibitors (ICIs) for clinical use in Asians are not as effective as in Caucasians [46,175].

## 7. Resistance Mechanisms of Mucosal Melanoma

In general, tumor patients with resistance to available therapy are clinically divided into two groups. One group of patients who exhibit primary resistance to therapeutic options and therefore do not benefit from the initial therapy, while the other group of patients who initially benefit from the offered therapy and may develop acquired resistance over time [89,217]. Consequently, treated tumor cells undergo adaptive resistance to evade the antitumor activities of available therapeutics, and thereby become able to regrow and to disseminate to distant organs. The development of acquired resistance can be mediated by intrinsically or extrinsically dependent mechanisms.

Although the most frequently discussed resistance mechanisms of different melanoma subtypes are mainly based on the analysis of CM, data obtained from the analysis of CM provide insights that may contribute to the understanding of the possible mechanisms regulating the resistance of melanoma subtypes, including MM [218,219]. Because it is a rare melanoma subtype, it is crucial to recognize the unique characteristics of MM and its potential differences in resistance mechanisms to advance its treatments.

## 8. Mechanisms of Intrinsic Resistance in Melanoma

Intrinsic melanoma resistance to available treatment options is mediated by a variety of mechanisms, including activation of alternative signaling pathways such as the PI3K-AKT-mTOR pathway, epithelial–mesenchymal transition (EMT), loss of critical cell surface receptors or antigens, expression of drug efflux pumps, and significant alterations in the tumor microenvironment [220,221]. All these cellular properties prevent cancer cells from responding to treatments even before the treatment has been initiated [160,222,223].

Since MM presents a significant clinical challenge due to its aggressive nature and limited treatment options, immunotherapy has emerged as a promising strategy for MM, with a particular focus on immune checkpoint inhibitors such as PD-1 and CTLA-4 inhibitors [160,162]. These inhibitors have shown significant efficacy by harnessing the body’s immune response against MM [224,225]. Adoptive cell transfer (ACT), antiangiogenic therapy and combination therapies have also attracted attention due to their potential in MM treatment [117,160]. In ACT, T cells are modified to specifically attack melanoma cells and show promising antitumor activity. While antiangiogenic therapy inhibits tumor growth through angiogenesis-dependent inhibition, the combination of immune checkpoint inhibitors with targeted therapies has been proposed as a versatile approach to overcome treatment resistance [162,226]. The mechanisms that regulate tumor cell intrinsic resistance to immunotherapy are mediated by genetic and molecular alterations in tumor cells that impair the infiltration and function of immune cells and thereby cause resistance to immunotherapies [227,228].

The resistance of cancer cells to immunotherapy is attributed mainly to the low antigen expression [229,230,231]. Furthermore, impairments in the mechanisms responsible for antigen processing and presentation, upregulation of constitutive PD-L1, deficiency of tumor-specific antigens and antigenic mutations, disruption of signaling pathways, genetic exclusion of T cells, and alterations in immune escape mechanisms may contribute significantly to evasion of immune responses, as widely reported [232,233]. Furthermore, mechanisms whose functional regulation is mediated by the E-box transcription factors TCF4 and BRD4, the transcriptional and epigenetic regulators, such as the cytoprotective enzyme heme oxygenase-1 (HO-1), the loss of Kelch-like ECH-associated protein 1 (KEAP1) and the loss of E-cadherin, are described as resistance-driving factors that can inhibit the blockades of immune checkpoints and subsequently trigger melanoma resistance to immunotherapy [234,235,236].

Tumor resistance to immunotherapy can arise from impaired antigen presenting processes. Thus, targeting CTLA-4 and PD-L1 with their specific therapeutics is crucial to promote T cell-directed immune reactivation against cancer [237,238]. However, to increase the effectiveness of immunotherapy, the T cells present in the tumor microenvironment must be able to recognize cancer cells [239,240]. To that end, antigen presentation is essential to stimulate T cells to recognize tumor cells. Therefore, significant defects in components of the tumor antigen-presenting machinery can prevent T cells from recognizing cancer cells and thereby allow tumor cells to evade the immune system [241,242].

Of note, one of a noteworthy components of tumor antigen-presenting machinery, the human leukocyte antigen class I (HLA-I) is a noteworthy component of the tumor antigen-presenting machinery is HLA-I, whose expression level determines the resistance and the response of tumor cells to immune checkpoint blockade inhibitors [56,227,228,243]. Suppression of HLA-I antigen processing and presentation may therefore play a significant role in primary resistance to anti-CTLA-4 and anti-PD-1 therapy. Consequently, the development of a treatment option that can reverse HLA-I suppression could overcome resistance to immune checkpoint blockades in MM. In addition to HLA-1, β-2 microglobulin (β-2M) is another essential element of the antigen presentation machinery involved in MHC class I antigen presentation [56,244]. Loss of β-2M can lead to loss of MHC class 1 and compatible presentation of tumor antigens, which in turn hinders effective anti-tumor reactions and contributes to immune evasion and therapy resistance [245,246]. Therefore, B2M deficiency in patients significantly leads to the development of MM resistance to both anti-CTLA-4 and anti-PD-L1 therapy [247,248].

The critical role of interferon (IFN) in the antigen processing-and-presentation machinery, is mediated by the upregulation of both MHC-1and MHC-2. However, the role of IFN is becoming increasingly important, especially in the context of anti-PD-1 therapy, since the initial response to ICIs is associated with a pre-existing IFN-mediated immune activation, which includes the expression of MHC-2 in metastatic tumors [249,250]. Although the role of IFN-γ in the induction of PD-L1 expression has also been reported [251,252], the role of IFN-γ during the treatment with immunotherapy is controversial [253,254]. While disruption of the IFN signaling pathway in melanoma negatively affects tumor response to anti-CTLA-4 therapy [255,256], extensive IFN-γ exposure of melanoma cells increased their resistance to radiotherapy and anti-CTLA-4 treatment [27,257]. However, these contrasting observations might be because IFN-γ-induced expression of PD-L1 leads to the suppression of T cells, which in turn leads to the development of adaptive resistance. While prolonged IFN-γ exposure can trigger adaptive resistance via the STAT1 signaling pathway, independent of its effect on PD-L1 expression. Taken together, IFN is expected to play a multifaceted role in regulating melanoma resistance to ICIs and contribute to the development of primary, adaptive, and acquired resistance in melanoma.

The involvement of signaling pathways in the development of intrinsic resistance to immunotherapy has been reported in several studies. One of the most common signaling pathways is the mitogen-activated protein kinase (MAPK) pathway. This pathway can negatively affect the anti-tumor activity by regulation of the production of VEGF and IL-8 cytokines [258,259]. The upregulation of these cytokines has been reported to result in the dysregulation in T-cell infiltration [62,260]. Also, the ability of MAPK to trigger the expression of PD-L1 in melanoma cells conferring resistance BRAF inhibitors has been reported [261,262], suggesting that the inhibition of MAPK pathway may overcome MM resistance to immunotherapy.

The loss of PTEN, the inhibitor of PI3K pathway, has been shown to contribute to the development melanoma resistance to immunotherapy [263,264,265,266]. While the activation of PI3K pathway by the loss of PTEN has been shown to decrease T cell infiltration in tumor via mechanisms mediated by secretion of suppressive cytokines such as CCL2 and VEGF, and inhibition of autophagic machinery [265,266].

Also, the promotion of transition from epithelial to mesenchymal states via mechanism mediated by the transcription factor, ZEB1 leading to immune escape by decreasing CD8+ T-cell accumulation in melanoma [267,268]. However, the reduction of CD8+ T cell infiltration into tumors by downregulating CD8+ T cell-attracting chemokines and cytokines such as CXCL10, CCL3, CCL4, IFN- and TNF- was a positive regulator of anti-PD1 therapy [269,270].

As is known, the tumor neoantigens produced by tumor cells are different antigens that arise from specific genetic mutations in cancer cells [271,272]. However, the absence of tumor antigens or the lack of their mutation increases the resistance of tumor cells and is the main cause of tumor cell resistance to ICI immunotherapy [248,273]. Accordingly, frequent mutations of tumor antigens facilitate the recognition of tumor cells as foreign cells by the immune system and trigger an immune reaction against the tumors [217,271]. However, it may happen that tumor cells do not possess these tumor antigens or that they can only present them to a limited extent on the cell surface [241,242]. As a result, melanomas may experience an ineffective T cell response and subsequently resistance to T cell-based immunotherapies such as anti-PD-L1 and anti-CTLA-4. The intrinsic mechanisms involved in the development of tumor resistance to immune checkpoint blockades are shown in Figure 4.

## 9. Mechanisms of Extrinsic Resistance in Melanoma

The mechanisms that regulate the extrinsic resistance of tumor cells to immunotherapy include factors external to molecular processing within tumor cells [217,227]. However, the most common external factors include immunosuppressive cells, hypoxia, inhibitory receptors, and phenotype switching, which can trigger resistance to immunotherapy [268,274].

The interaction between components of the tumor microenvironment (TME), particularly immunosuppressive cells, inhibitory receptors, and exosomes, is the main factor that enables cancer cells to evade immune attacks through a mechanism mediated by the enhancement of the immunosuppressive environment [275,276]. Regulatory T cells (Treg cells) are a T cell type functionally responsible for suppressing the immune system to prevent excessive inflammation and autoimmunity [277,278]. Thus, Treg cells have the potential to dampen the immune response via a mechanism mediated by the production of cytokines such as interleukin (IL)-10 and -35 [279,280].

In addition to their ability to suppress effector T cells, Treg cells can also stimulate tumor-infiltrating macrophages to produce B7-H molecules [281,282]. As a result, the interaction of these molecules with their corresponding ligands induces immune tolerance via a mechanism mediated by the attenuation of the T cell response [283,284]. Also, the interaction of Treg cells with other immunosuppressive cells in the TME through the secretion of various cytokines, resulting in an immunosuppressive microenvironment [285,286]. Furthermore, the ability of Treg [285,287]. The mechanisms by which apoptotic Treg cells induce an immunosuppressive microenvironment are attributed to the ability of apoptotic Treg cells to release large amounts of ATP and convert it into immunosuppressive adenosine. This explanation could be one of the possible mechanisms by which Treg cells develop resistance to PD-L1 therapy. Also, the involvement of hypoxia-induced Treg cell apoptosis may contribute to the evolution of immune evasion mechanism in TME leading to development of resistance to ICIs [288,289]. The involvement of hypoxia-induced Treg cell apoptosis may also contribute to the development of an immune escape mechanism in TME, leading to the development of resistance to ICIs. The role of myeloid-derived suppressor cells in regulating melanoma resistance and response to immunotherapy has been reported [286,290]. These myeloid-derived suppressor cells are a group of myeloid cells with immunosuppressive potential [291,292]. One of the most common mechanisms by which myeloid cells mediate their immunosuppressive activity is based on the increased expression of nitric oxide (NO) and arginase (Arg)-1. While the main function of Arg-1 is to deplete L-arginine, which is essential for T cell function, the increased expression of No serves to inhibit T cell proliferation [293,294]. Consequently, myeloid-derived suppressor cells may play an important role in regulating T cell dysfunction and attenuating response to immunotherapies, including ICIs [295,296].

Although the high concentrations of myeloid suppressive cells in melanoma patients did not provide any therapeutic benefit after treatment with the immunotherapy ipilimumab [297,298], prolonged survival and objective clinical responses were observed when treating patients with advanced melanoma with ipilimumab, particularly in patients with low CD33+CD11b+HLA-DR myeloid-derived suppressor cells [299,300]. Furthermore, increased myeloid suppressor cell population and lack of functional T cells have been reported to impair the efficacy of ICIs [295,296].

The interaction between the cellular components of TME determines their molecular properties [275,301]. Thus, TME may have the ability to alter the properties of cancer cells as well as the expression profile of the cells in their microenvironment over time [302,303]. The analysis of numerus tumors derived from the cancer genome identified a subset of immunosuppressive Tregs that invade the melanoma microenvironment [286,304]. These Tregs are characterized by their ability to dampen the immune response through the production of IL-10 and IL-35 cytokines [305,306]. In addition to their ability to suppress effector T cells, Tregs have been found to stimulate tumor-infiltrating macrophages to produce B7-H molecules [281,282]. However, the interaction of these molecules with their ligands was found to contribute to immune tolerance by dampening the T cell response [307,308].

Also, as one of the most important cellular components of the TME, cancer-associated fibroblasts (CAFs) have been widely reported to contribute to the regulation of tumor invasiveness [309,310]. Therefore, TME, where CAFs are present in high abundance, is usually associated with an invasive phenotype of melanoma cells [74,311]. Of note, this invasive phenotype of melanoma cells is characterized by microphthalmia-associated transcription factor (MITF) low/AXL high expression [312,313]. However, phenotype switching is a mechanism that allows melanoma cells to switch between different cell states [314,315]. Therefore, resistance of melanoma to immunotherapies such as PD-1 inhibitors is associated with de-differentiation of melanoma [274,316]. Therefore, most patients who responded to anti-PD-1 therapy had differentiated gene signatures characterized by high proliferation and low invasiveness [317,318]. It is well documented that Tregs within the TME underwent programmed cell death, and thus, apoptotic Tregs can more effectively suppress T cell activation, mainly by releasing high concentrations of ATP and converting it into immunosuppressive adenosine using specific enzymes [319,320].

The role of tumor associated macrophages (TAMs) belongs to the cellular components of TME [275,321]. Although TAMs exhibit distinct phenotypes, including M1-like and M2-like features, only the M2-like phenotype is associated with immunosuppressive functions [58,322]. Targeting the scavenger receptor MARCO on TAMs by monoclonal antibodies has been reported to reduce the presence of M2TAMs and improve the efficacy of anti-CTLA-4 antibody therapy [323,324]. However, several studies have demonstrated the mechanisms by which TAMs promote resistance to immune checkpoint therapy [325,326]. One of these mechanisms is mediated by the secretion of immunosuppressive molecules such as tumor growth factor (TGF) and prostaglandin E2 (PGE2) [327,328]. As a result, both TGF and PGE2 can inhibit the activity of cytotoxic cells and thus enhance Treg expansion, which in turn attenuates the anti-tumor immune response [329,330]. In addition, TAMs have the potential to release immune checkpoint proteins such as PD-L1, which directly suppress T cell activity [80,331]. However, it has been reported that PD-L1 expression in TAMs is associated with the development of resistance to PD-1/PD-L1 blockade-dependent therapy [332,333]. The mechanisms of the extrinsic resistance of melanoma to ICIs are outlined in Figure 5.

## 10. Conclusions

In contrast to cutaneous melanoma (CM), mucosal melanoma (MM) is characterized by a lower response to already proven treatment options for different tumor types including CM. However, the discovery of immune checkpoint inhibitors (ICIs) has led to significant improvements in immunotherapy. Compared to the available anti-cancer agents, immunotherapy is the best treatment option for MM. Although ICIs are effective in both adjuvant and primary treatment, especially in melanoma subtypes such as MM, many patients have developed resistance to ICIs and have not been able to benefit from treatment. Due to the common origin of melanomas, some resistance mechanisms may also be common between CM and MM. However, it is important to remember that MM has unique clinical and molecular features that may lead to resistance mechanisms not observed in CM. In summary, it is important to thoroughly investigate the factors contributing to resistance to ICI therapy and to develop effective combination treatments to overcome the obstacles of MM resistance.

## Figures and Tables

**Figure 1 cells-14-01884-f001:**
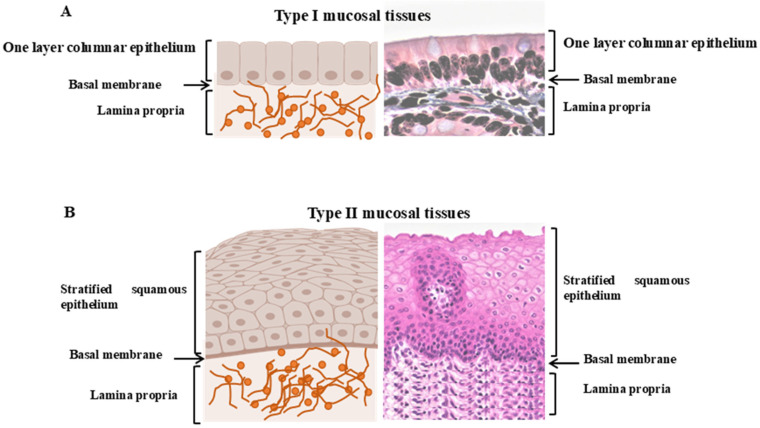
Structures of mucous membrane surfaces in the human body. (**A**) Type I mucosal tissues consist of an epithelial cell layer covered by a protective mucous layer maintained by cells such as goblet cells. This type of mucosal tissues lines the intestines, respiratory tract, and upper female reproductive tract. They act as a physical and chemical barrier, interface to the external environment and have unique immune components such as a mucous layer with embedded immune cells and the local production of secretory IgA (sIgA). (**B**) Type II mucosal tissue consists of stratified squamous epithelial layers and forms the mucosal surfaces of the mouth and lower female reproductive tract. They are characterized by specific immune cells and pathways that are essential for the activation of innate lymphoid cells type 2 and the induction of inflammatory responses by submucosal dendritic cells. Created on 20 November 2025 by BioRender.com.

**Figure 2 cells-14-01884-f002:**
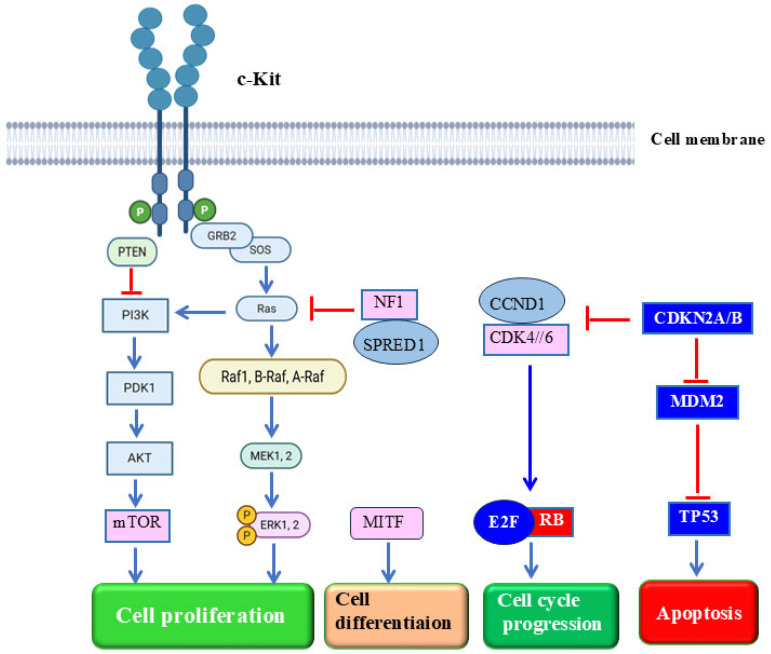
Main signaling pathways and their functional role in MM. BRAF: B-Raf proto-oncogene; CDK4/6: Cyclin-dependent kinase 4/6; CCND1: Cyclin D1; CDKN2A/B: Cyclin-dependent kinase inhibitor of kinase 2A/B; E2F: Transcription factor 2; ERK: Extracellular related kinase; KIT: receptor tyrosine kinase; MDM2: mouse double minute 2 p53 binding protein homolog; MITF, Microphthalmia-associated transcription factor; MEK:MAP kinase; mTOR: mammalian target of rapamycin; NF1: neurofibromin 1; NRAS: neuroblastoma RAS viral oncogene homolog; PI3K: Phosphatidylinositol 3 kinase; PTEN: Phosphatase and tensin homolog; RB1: retinoblastoma 1; SPRED1: sprout-related EVH1 domain containing protein 1; TP53: tumor suppressor protein. Created by BioRender.com.

**Figure 3 cells-14-01884-f003:**
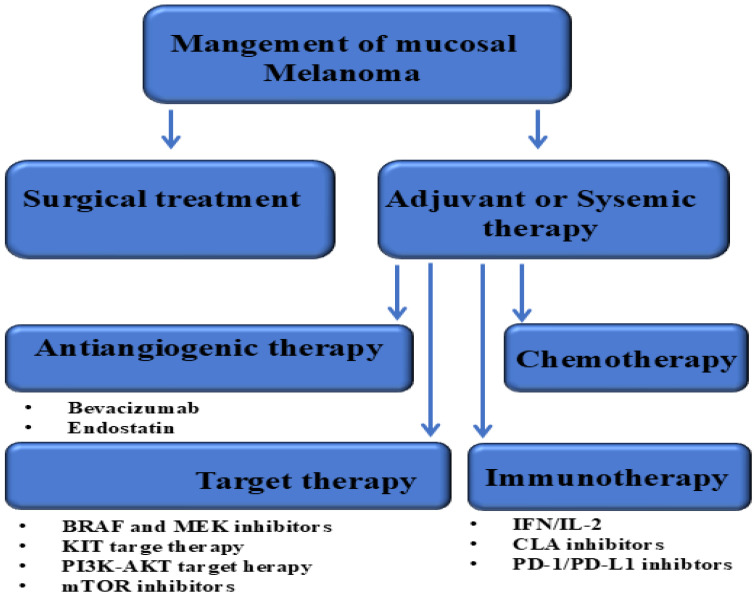
Treatment options of mucosal melanoma.

**Figure 4 cells-14-01884-f004:**
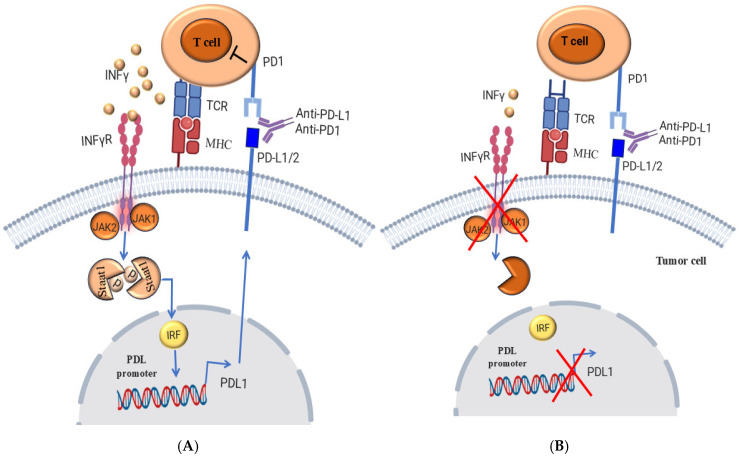
(**A**) An anti-tumor immune response is essential for effective blockade of immune checkpoints. Tumor-reactive T cells that recognize tumor neoantigens in MHC class I or class II-dependent interferon-γ (IFNγ). As a result, IFNγ enhances the activation of the IFNγ receptor (IFNγR), leading to JAK1/JAK2-dependent phosphorylation and dimerization of state 1, which in turn triggers the adaptive expression of programmed cell death ligand 1 (PD-L1) on the surface of tumor cells, which can negatively regulate the antitumor T cell response. (**B**) Tumor-specific T cells interact with MHC-antigens on the surface of tumor cells, leading to the release of IFNγ. Due to genetic deficiencies in IFNγ signaling in tumor cells, the IFNγ signal cannot be further transduced to enhance adaptive PD-L1 expression. As a result, blockade of the PD 1-PD-L1 immune checkpoint becomes ineffective in triggering an anti-tumor immune response. As, consequence, the tumor cells develop resistance to immunotherapy. Created by BioRender.com.

**Figure 5 cells-14-01884-f005:**
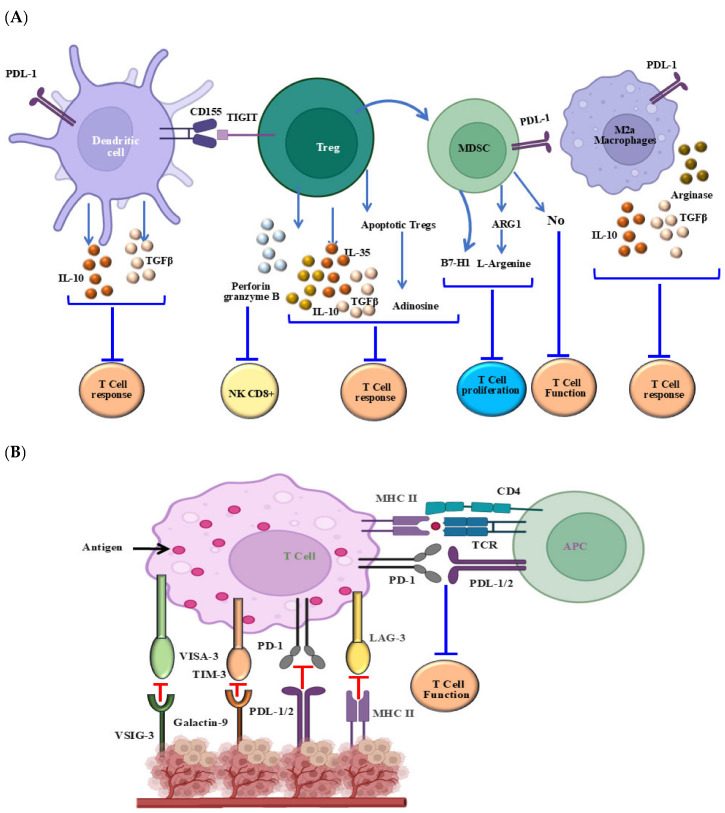
Extrinsic mechanisms involved in the regulation of tumor resistance to immune checkpoint blockade. (**A**) Immune-suppressive cell components including regulatory T cells (Treg), tumor-associated macrophages (TAMs), regulatory dendritic cells (DC), and myeloid-derived suppressor cells (MDSC). (**B**) Inhibitory receptors (PD L1, LAG-3, TIM-3, and VISTA) and their biological consequences. (**C**) Mechanisms of phenotype switching. (**D**) Hypoxic tumor microenvironment-mediated effects. **Abbreviations**: LAG-3, lymphocyte-activation gene 3; PDL-1, programmed death ligand 1; TIM-3, T-cell immunoglobulin and mucin domain 3; VISTA, V-domain I g suppressor of T cell activation; HIF-1α, hypoxia-inducible factor 1-alpha; MITF, microphthalmia-associated transcription factor. Created by BioRender.com.

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
