# Peer review of "Mucosal Melanoma: Mechanisms of Its Etiology, Progression, Resistance and Therapy"

_cells, 2025, doi:10.3390/cells14231884_

Round 1
Reviewer 1 Report
Comments and Suggestions for Authors
In this narrative review, the authors reported clear evidens regarding mucose melnoma and discussed resistance of the drugs, prognosis and life span prolongtion. The aim of the study and the initial hypothesis are clearly articulated and well defined. The authors present ideas that are logically structured, conceptually sound, and convincingly supported by relevant evidence. The manuscript is written in a clear, coherent, and accessible manner, demonstrating a high standard of academic writing. The study is both relevant and timely, effectively situated within the context of the existing literature. The authors successfully identify key gaps in current knowledge and position their research as a meaningful contribution to addressing these deficiencies. Their interpretation of statistical analyses is appropriate and insightful, and the discussion thoughtfully integrates their findings with those of recent studies in the field. Furthermore, the manuscript offers a well-reasoned and comprehensive conclusion that highlights the significance of the results while acknowledging potential limitations and suggesting directions for future research. Overall, the work demonstrates academic rigor, originality, and clarity, making it a valuable addition to the body of knowledge on the topic. Although the findings are impressive, I wouild like to make the following comments:
- Immuno-and target therapies are likely to be reported separetely in close link with a presence of appropriate mutations. The diagramm can be modified taking into consideration this issue.
- Conclusive part: This section should be more concrete and focused on practical significance. Perhaps it should be shortened a bit.
Author Response
Authors’ response to reviewer 1
In this narrative review, the authors reported clear evidens regarding mucose melnoma and discussed resistance of the drugs, prognosis and life span prolongtion. The aim of the study and the initial hypothesis are clearly articulated and well defined. The authors present ideas that are logically structured, conceptually sound, and convincingly supported by relevant evidence. The manuscript is written in a clear, coherent, and accessible manner, demonstrating a high standard of academic writing. The study is both relevant and timely, effectively situated within the context of the existing literature. The authors successfully identify key gaps in current knowledge and position their research as a meaningful contribution to addressing these deficiencies. Their interpretation of statistical analyses is appropriate and insightful, and the discussion thoughtfully integrates their findings with those of recent studies in the field. Furthermore, the manuscript offers a well-reasoned and comprehensive conclusion that highlights the significance of the results while acknowledging potential limitations and suggesting directions for future research. Overall, the work demonstrates academic rigor, originality, and clarity, making it a valuable addition to the body of knowledge on the topic. Although the findings are impressive, I wouild like to make the following comments:
Comment 1: Immuno-and target therapies are likely to be reported separetely in close link with the presence of appropriate mutations. The diagram can be modified taking into consideration this issue.
Authors’ response: Thank you very much for comment. Accordingly chapters of immune-and targeted therapy were structured and reported separately. And kept the diagram in the introduction of the chapter as a general concept for the whole chapter
……
See lines ; 417 to 593; the modifications of chapter of the therapeutic options
Therapeutic option of mucosal melanoma
MM exhibits heterogeneity on several levels, including genetic, molecular, and phenotypic differences between different tumors as well as within a single tumor [(113) (1)]. This is reflected in the diverse genetic mutations, which differ from those of cutaneous CM and vary depending on the tissue of origin, as well as in the different cellular subpopulations with varying metastatic potential and metabolic profiles [(151) (152)]. Consequently, the frequent occurrence of heterogeneity between different subgroups of mucosal melanoma, as well as within a single tumor, complicates treatment and underlines the need for personalized strategies based on individual tumor analyses [(40, 153)].
Although numerous technological developments in surgery, radiotherapy, and systemic treatment modalities, no increased survival benefit was observed in MM(154, 155) [150, 151]. The heterogeneity of MM at different sites presents a challenge for treatment.
However, MM poses a significant challenge in achieving negative resection margins due to its advanced stage at diagnosis, frequent lymph node involvement, complex anatomical locations and multifocal lesions (122, 156). Consequently, surgical treatment of MM is usually associated with a recurrence rate of 50% to 90%(50, 117, 148). Thus, the treatment of MM requires a multidisciplinary approach particularly for complex anatomical locations and multifocal lesions(26, 157). However, neoadjuvant and adjuvant therapies are expected to improve outcomes in resectable MM(158, 159). While the most common treatment options for advanced or metastatic disease are radiotherapy, immunotherapy and targeted therapy(41, 160). The therapeutic management of anorectal melanoma is outlined in figure 3.
Figure 3. Treatment options of mucosal melanoma.
Immune therapy
Although the first attempts to use the immune system to fight cancer were only made towards the end of the 19th century, significant progress has been made first in recent years (161). However, the elucidation of the immune mechanisms of multiple myeloma paved the way for the accelerated development of innovative immunotherapeutic strategies. Immune checkpoint inhibitors (ICIs) have proven to be promising agents capable of disrupting immune checkpoint interactions, thereby effectively activating the host's immune system leading to the destruction of tumor cells(148, 149). In parallel with ICIs, other forms of therapy such as adoptive cell therapy (ACT), vascular endothelial growth factor (VEGF) inhibitors and combination therapies have also shown a strong anti-timor effect (150, 154). Clinical trials investigating the efficacy of ICIs include anti-CTLA-4 and anti-PD-1 therapies alone or in combination.
While immunotherapy, especially with PD-1 inhibitors, is the more effective therapeutic approach for metastatic and surgically unresectable MM (162, 163). Even though the treatment responses are lower in MM than in CM(9, 10) (41).
Clinical studies of VEGF inhibitors in MM predominantly involve combinations with PD-1 inhibitors and show promising results such as objective response rates (ORR) of about 40-45% in early studies (122, 157).
PD-1 inhibitors form a pharmacological subclass of therapeutics that specifically target the immune checkpoint protein PD-1 and thereby inhibit the interaction between PD-1 and its ligands (157, 158). These remarkable advances in immunotherapy herald a new era in cancer treatment, including the treatment of MM (104, 155). This inhibition restores and enhances the immune response of activated T cells against tumor cells(160, 162). Importantly, PD-1 inhibitors have demonstrated considerable efficacy in patients with MM, as evidenced by several clinical trials that confirm their remarkable performance in MM therapy (163, 164). These studies highlight the favorable and sustained results of PD-1 inhibitor monotherapy and offer survival benefits for patients with advanced and metastatic melanoma(165, 166). In addition, a study on the efficacy of anti-PD-1 therapy in patients with head and neck mucosal melanoma (HNMM) was conducted, in which the expression of PD-L1 and PD-1 in tumor tissue samples was examined (167, 168). Although a low PD-L1 expression level was observed in HHMM samples, treatment with anti-PD-1 therapy did not lead to clinical success in these patients (147, 155). This is evidence that highlights the challenges and limitations of checkpoint inhibitors in the treatment of advanced MM.
As is reported, CTLA-4 inhibitors mediate their effect mainly through the specific inhibition of the CTLA-4 protein expressed on the cell surface, the essential modulator of T-cell function (169, 170). These inhibitors mediate their antitumor activity by inhibiting the interaction between CTLA-4 and its ligands (B7-1 and B7-2), which in turn suppresses inhibitory signals to T cells and leads to enhanced antitumor immune responses(136) (171). Although initial clinical studies have consistently shown remarkable therapeutic efficacy of CTLA-4 inhibitors in patients with MM, Patients with advanced mucosal melanoma who had failed nivolumab therapy (104, 155)[. In contrast, treatment with ipilimumab as an adjuvant therapy has been reported to offer additional benefits in terms of progression-free survival in patients with advanced, nivolumab-resistant mucosal melanoma (155, 172-175).
The PD-1/PD-L1 and CTLA-4 signaling pathways are important immune checkpoint pathways with different mechanisms of action in cancer therapy (171, 176). Therefore, the combination of PD-1/PD-L1 and CTLA-4 inhibitors may be more effective than monotherapy in MM (136, 177). Although increasing evidence suggests that combining ICIs enhances the immune response and improves survival, a frequent increase in the risk of serious immune-related adverse events has been observed (178, 179). The combination therapy, which often includes drugs such as nivolumab (anti-PD-1) and ipilimumab (anti-CTLA-4), works by simultaneously 'priming' and 'enhancing' T-cell responses to release an antitumor potential against the tumor(33, 180). Although mucosal melanomas generally have lower response rates than other melanoma subtypes, combination therapy is an important area for future research to overcome resistance and control toxicity.
Of note, the highest 5-year OS rate among therapies for advanced melanoma was observed with combined anti-CTLA-4 and anti-PD-1 checkpoint inhibition, which is most strongly associated with the highest incidence of immune-related adverse events (176, 181). Data from comparative clinical trials of monotherapy and combination therapy with nivolumab and ipilimumab showed that combination therapy offers advantages.
Clinical studies of VEGF inhibitors in MM predominantly involve combinations with PD-1 inhibitors and show promising results such as objective response rates (ORR) of about 40-45% in early studies (123, 182). These studies use drugs such as axitinib (a VEGFR inhibitor) and nivolumab or toripalimab (PD-1 inhibitors). It has been reported that combining anti-VEGF therapy with PD-1 blockade could be more effective than single-drug immunotherapy or chemotherapy alone (182, 183). Some studies have also examined chemotherapy in combination with bevacizumab (a VEGF inhibitor) (182, 184).
The highest 5-year OS rate among therapies for advanced melanoma was observed with combined anti-CTLA-4 and anti-PD-1 checkpoint inhibition, which is most strongly associated with the highest incidence of immune-related adverse events(176, 181). Data from comparative clinical trials of monotherapy and combination therapy with nivolumab and ipilimumab showed that combination therapy offers advantages over monotherapy, as demonstrated by the significant increase of the median PFS and objective response rates (ORRs) (185, 186).
Although the median response time was similar in MM and CM, apart from the type of treatment, patients receiving nivolumab monotherapy showed lower median PFS of 3.0 months in MM and 6.2 months in CM, with ORRs of 23.3% and 40.9%, respectively (178, 179). In addition, patients who received ipilimumab monotherapy had a median PFS of 2.7 months for MM and 3.9 months for CM (178, 179) . While patients receiving nivolumab in combination with ipilimumab had a better median PFS of 5.9 months for MM and 11.7 months for CM, with ORRs of 37.1% and 60.4%, respectively (180, 187).
In summary, although the responses of MM to ICIs have been observed in several studies, the efficacy of ICIs in MM is relatively suboptimal compared to CM and other melanoma subtypes. However, ICIs remain the best therapeutic option for MM.
Due to its anatomical location, MM is characterized by a high tendency to vascular invasion, the mechanism that can relatively enhance its sensitivity to antiangiogenic agents (182, 188, 189). Consequently, combination therapy based on antiangiogenesis could offer a therapeutic opportunity.
The combination of the toripalimab, a humanized anti–PD-1 monoclonal antibody, in combination with the inhibitor of vascular endothelial growth factor (VEGF) receptor, axitinib for advanced MM showed an ORR of 48.3% and a disease control rate (DCR) of 86.2%(182, 190). In addition, the median duration of response (DoR) was 13.7 months, the median PFS was 7.5 months, and the median OS was 20.7 months, as shown in a Phase 1b study (183, 191).
While in phase II trial evaluation of atezolizumab, the anti-PD-1 monoclonal antibody with the VEGF monoclonal antibody, bevacizumab in advanced MM revealed an ORR of 36.4%, with a median PFS of 5.2 months and a DCR of 59.1%(46, 192).
The evaluation of the efficacy and safety of the humanized anti-PD-1 antibody camrelizumab in combination with the VEGFR tyrosine kinase inhibitor apatinib in advanced MM showed an ORR of 42.9%, a DCR of 82.1%, and a median PFS of 8.05 months(193, 194).
In addition, the combination of chemotherapy with antiangiogenic agents has been approved due to its efficacy in the treatment of unresectable or advanced MM(195, 196).
Preferentially expressed Antigen in Melanoma (PRAME), a multifunctional cancer testicular antigen expressed in normal and neoplastic tissue, has proven to be a useful diagnostic tool in the differential diagnosis between benign and malignant melanocytic lesions (111) (112). In addition to be a promising target for immunotherapy, PRAME was approved as a diagnostic marker for subgroups of mucosal melanoma, including urological, gynecological, and head and neck mucosal melanomas, due to its reliability [(113) (114)]. Of note, the differential expression of PRAME in different subtypes of mucosal melanoma therefore makes it useful to distinguish malignant melanomas from their benign counterparts (112) (115)
Radiotherapy
Adjuvant radiotherapy is a postoperative treatment for mucosal melanoma that is often recommended to reduce the risk of local and regional recurrence, especially in patients with high-risk factors such as positive resection margins, advanced stage, or multiple positive lymph nodes(165, 166) Although adjuvant radiotherapy can improve local control, its impact on overall survival in MM remains controversial and should perhaps be considered in conjunction with modern systemic therapies (197) (198).
Radiotherapy is often used as an adjuvant treatment in the treatment of MM, especially for postoperative lesion control or when surgery is not possible (197) (198). Although radiation therapy is effective in killing cancer cells, it can damage surrounding normal tissue [164, 165]. Chemotherapy is a systemic treatment option for advanced or metastatic MM that aims to relieve symptoms and slow tumor growth [166, 167].
It has been shown that radiotherapy promotes the infiltration of T cells into the tumor microenvironment and acts synergistically with the immune system by inducing the release of tumor-associated antigens [(199) (200)]. This can lead to the local death of tumor cells and simultaneously trigger an abscopal effect, in which immune cells attack distant, non-irradiated tumor sites. [(201) (202)]. However, the efficacy of pembrolizumab in combination with hypofractionated radiotherapy in patients with mucosal melanoma is being investigated in the clinical trial [(203) (204)]. Even untreated patients with primary sinonasal mucosal melanoma showed promising survival benefits from the combination of surgery, radiotherapy and immunotherapy, as demonstrated in a phase II trial of nivolumab and axitinib [(205) (206)]. To that end, the combination therapy significantly improved progression free survival (PFS) and overall survival (OS)(207, 208). This was demonstrated by a Phase III clinical trial in patients with advanced BRAF V600E-mutated melanoma, based on the investigation of the safety and efficacy of combination therapy with BRAF and a MEK inhibitors(169, 170).
Targeted therapies
Targeted therapies, especially for specific mutations such as BRAF or KIT have been suggested for MM treatment(128, 164).
In contrast to CM, MM has a higher KIT mutation rate, which may improve its response to KIT inhibitors compared to other melanomas(13, 136). There are no clinical trials specifically investigating KIT inhibitors in MM, most studies included a significant proportion of non-CM subtypes, with MM accounting for 46% to 71% of included cases(171, 177) [175, 176]. Although imatinib represents a treatment option for patients with KIT mutations, its use is often limited to clinical trials (209) (210). In addition, many other targeted therapies are being investigated in clinical trials, sometimes in combination with other treatments such as immunotherapy (211) (212). The development of therapeutic approaches, including neoadjuvant and adjuvant therapies, is critically evaluated for the treatment of melanoma subtypes including MM (117) (213).
The BRAF V600E mutation rate of MM is estimated at approximately 12% and could therefore benefit from combination therapy with BRAF and a MEK inhibitor (117, 214, 215).
The most frequently used targeted therapies for mucosal melanomas are BRAF and MEK inhibitors, which can be administered in combination, e.g., brafenib plus tramatinib, especially in patients with a BRAF V600 mutation (117) (160). However, if the melanoma has a C-KIT mutation, targeted therapies such as imatinib or nilotinib can be used (216) (40).
In contrast to CM, MM has a higher KIT mutation rate, which may improve its response to KIT inhibitors compared to other melanomas (13, 136, 171). Although there are no clinical trials specifically investigating KIT inhibitors in MM, most studies included a significant proportion of non-CM subtypes, with MM accounting for 46% to 71% of included cases (171, 177).
The loss of phosphoinositide-dependent kinase-1 (PDPK1) showed clinical relevance for the therapeutic efficacy of MEKis in melanoma, as demonstrated by clinical studies (172, 208). However, the synergistic effect of PDPK1 depletion and MEKi was further investigated in melanoma cell lines with NRAS-mutation(172, 173). The clinical relevance of the combination of PDPK1 and MEK inhibitors was also demonstrated in the NRAS-mutated xenograft model, indicating the efficiency of the combinatorial strategy for the treatment of melanoma patients with NRAS mutation (172, 174). Although immune checkpoint inhibitors (ICIs) for clinical use in Asians are not as effective as in Caucasians (46, 175).
Comment 2: Conclusive part: This section should be more concrete and focused on practical significance. Perhaps it should be shortened a bit.
Authors’ response: Thank you very much for your comment. Accordingly, we modified the conclusion as required
See lines: 806-815
In contrast to cutaneous melanoma (CM), mucosal melanoma (MM) is characterized by a lower response to already proven treatment options for different tumor types including CM. However, the discovery of immune checkpoint inhibitors (ICIs) has led to significant improvements in immunotherapy. Compared to the available anti-cancer agents, immunotherapy is the best treatment option of MM. Although ICIs are effective in both adjuvant and primary treatment, especially in melanoma subtypes such as MM, many patients have developed resistance to ICIs and have not been able to benefit from treatment. Due to the common origin of melanomas, some resistance mechanisms may also be common between CM and MM. However, it is important to remember that MM has unique clinical and molecular features that may lead to resistance mechanisms not observed in CM. In summary, it is important to thoroughly investigate the factors contributing to resistance to ICI therapy and to develop effective combination treatments to overcome the obstacles of MM resistance.

Reviewer 2 Report
Comments and Suggestions for Authors
Thank you to the Editor for allowing me to review this interesting Manuscript. I recommend to accept it after major revisions, as follows:
1- The manuscript is detailed and comprehensive. However, some issue deserves further details, especially regarding molecular landscape:
- A) Example-1: the authors affirm that BRAF and RAS mutations are more characteristic of CM rather than MM (true). However, RAS/BRAF-axis mutations, especially NRAS, are the the most frequent ones in sinonasal melanoma, the most frequent subtype of H&N-MM (PMID: 35978013). This point needs to be clarified in the Manuscript; otherwise, there is a risk of providing incorrect information;
- B) Example-2: It is widely recognized that MM of different sites (urological, gynecological, and H&N) are molecularly distinct diseases. Additionally, subgroups from the same site (e.g., oral cavity vs. sinonasal) exhibit different molecular landscapes. Please specify this point;
2 – Please, describe the limitats affecting the staging (pT) of MM;
3 – Please, specify and describe PRAME data in MM and in the different subgroups (urological, gynecological, and H&N);
4 - In MM-H&N (and probably in other MMs), the particular anatomical location does not allow for radical surgery or imply an high number of residual tumor/relapses. Recently, some authors found that the molecular landscape of different samples collected from a single primary MM-H&N (excision of the primary tumor, residual tumor/relapses, and metastases) may differ (“molecular heterogeneity” – PMID: 40231352). This represents a crucial mechanism of resistance. Please discuss this point.
Author Response
Auhors’response to reviewer 2
Comment1: A) Example-1: the authors affirm that BRAF and RAS mutations are more characteristic of CM rather than MM (true). However, RAS/BRAF-axis mutations, especially NRAS, are the the most frequent ones in sinonasal melanoma, the most frequent subtype of H&N-MM (PMID: 35978013). This point needs to be clarified in the Manuscript; otherwise, there is a risk of providing incorrect information;
Authors’response: Thank you very much for your comment. Accordingly, we added a paragraph to the manuscript
See lines: 350-356; the following paragraph
Mutations in the RAS/BRAF axis system, especially NRAS, are common in sinonasal melanomas, often even more common than other common melanoma mutations such as BRAF (1) (2)]. Some studies suggest that RAS mutations, particularly in the NRAS gene, are the most common driver mutations in sinonasal melanomas [(1) (3)]. Although BRAF is the most frequently mutational oncogene in cutaneous melanoma, studies of sinonasal melanomas show a higher prevalence of NRAS mutations compared to BRAF mutations [(4) (5)].
Has been added to the main text.
Comment 2: B) Example-2: It is widely recognized that MM of different sites (urological, gynecological, and H&N) are molecularly distinct diseases. Additionally, subgroups from the same site (e.g., oral cavity vs. sinonasal) exhibit different molecular landscapes. Please specify this point;
Authors’response: Thank you very much for your comment. Accordingly, we added a paragraph to the manuscript
See lines: 338-349; the following paragraph:
Although MMs may share some common characteristics, as they all fall under the broader category of mucosal melanomas [(6) (7)], MMs of different sites, including those in the head and neck area as well as those of urological and gynecological mucosal origin, are molecularly different [(6) (7)]. They differ in their specific mutation patterns, such as the higher frequency of TP53 mutations and the absence of BRAF mutations in urethral melanoma compared to CM [(8) (9)]. They also differ in other genetic alterations such as KIT and NRAS mutations, which are known to affect MMs response to targeted therapies [(3) (10)]. For example, urethral melanomas exhibit a high frequency of TP53 mutations and a remarkable lack of BRAF mutations [[(8) (9)]. Although TP53 and KIT mutations are observed in gynecological melanomas, BRAF and NRAS mutations have been shown to occur in some cases [(11) (12)].
has been added to the main text
Comment 3: Please, describe the limitats affecting the staging (pT) of MM;
Authors’response: Thank you very much for your comment. As required, we described the most common factors that affect mucosal melanoma staging.
See lines :116-122; the following paragraph :
The main limitations affecting mucosal melanoma staging are attributed to the lack of a standardized, and specific system for mucosal melanoma staging [(13) (14)]. Other factors that complicate staging include the use of systems for other cancers, which may not be suitable for staging mucosal melanoma, the specific anatomical location of the tumor, its aggressive behavior, and the tendency for late diagnosis [(14, 15)]. Although there are ongoing efforts to develop specific risk-based tiered models, the lack of a generally accepted system represents a major limitation (13, 16)].
Has been added to the main text.
Comment 4: Please, specify and describe PRAME data in MM and in the different subgroups (urological, gynecological, and H&N);
Authors’response: Thank you very much for your comment. Accordingly, we added a paragraph describing the validity of PRAME in MM
See lines :301-309; the following paragraph :
Preferentially expressed Antigen in Melanoma (PRAME), a multifunctional cancer testicular antigen expressed in normal and neoplastic tissue, has proven to be a useful diagnostic tool in the differential diagnosis between benign and malignant melanocytic lesions (17) (18). In addition to be a promising target for immunotherapy, PRAME was approved as a diagnostic marker for subgroups of mucosal melanoma, including urological, gynecological, and head and neck mucosal melanomas, due to its reliability [(19) (20)]. The differential expression of PRAME in different subtypes of mucosal melanoma therefore makes it useful to distinguish malignant melanomas from their benign counterparts (18) (21)
Has been added to the main text.
Comment 5: In MM-H&N (and probably in other MMs), the particular anatomical location does not allow for radical surgery or imply an high number of residual tumor/relapses. Recently, some authors found that the molecular landscape of different samples collected from a single primary MM-H&N (excision of the primary tumor, residual tumor/relapses, and metastases) may differ (“molecular heterogeneity” – PMID: 40231352). This represents a crucial mechanism of resistance. Please discuss this point.
Authors’response: Thank you very much for your commen. Accordingly, we added a paragraph about heterogeneity of MM.
See lines: 418-426; the following paragraph:
MM exhibits heterogeneity on several levels, including genetic, molecular, and phenotypic differences between different tumors as well as within a single tumor [(19) (6)]. This is reflected in the diverse genetic mutations, which differ from those of cutaneous CM and vary depending on the tissue of origin, as well as in the different cellular subpopulations with varying metastatic potential and metabolic profiles [(22) (23)]. Consequently, the frequent occurrence of heterogeneity between different subgroups of mucosal melanoma, as well as within a single tumor, complicates treatment and underlines the need for personalized strategies based on individual tumor analyses [(24, 25)].
Has been added to the main text.
- Chlopek M, Lasota J, Thompson LDR, Szczepaniak M, Kuzniacka A, Hincza K, et al. Alterations in key signaling pathways in sinonasal tract melanoma. A molecular genetics and immunohistochemical study of 90 cases and comprehensive review of the literature. Mod Pathol. 2022;35(11):1609-17.
- Heidrich I, Rautmann C, Ly C, Khatri R, Kott J, Geidel G, et al. In-depth assessment of BRAF, NRAS, KRAS, EGFR, and PIK3CA mutations on cell-free DNA in the blood of melanoma patients receiving immune checkpoint inhibition. J Exp Clin Cancer Res. 2025;44(1):202.
- Cosgarea I, Ugurel S, Sucker A, Livingstone E, Zimmer L, Ziemer M, et al. Targeted next generation sequencing of mucosal melanomas identifies frequent NF1 and RAS mutations. Oncotarget. 2017;8(25):40683-92.
- Dumaz N, Jouenne F, Delyon J, Mourah S, Bensussan A, Lebbe C. Atypical BRAF and NRAS Mutations in Mucosal Melanoma. Cancers (Basel). 2019;11(8).
- Si L, Wang X, Guo J. Genotyping of mucosal melanoma. Chin Clin Oncol. 2014;3(3):34.
- Santeufemia DA, Palmieri G, Miolo G, Colombino M, Doro MG, Frogheri L, et al. Current Trends in Mucosal Melanomas: An Overview. Cancers (Basel). 2023;15(5).
- Sergi MC, Filoni E, Triggiano G, Cazzato G, Interno V, Porta C, et al. Mucosal Melanoma: Epidemiology, Clinical Features, and Treatment. Curr Oncol Rep. 2023;25(11):1247-58.
- Mano R, Hoeh B, DiNatale RG, Sanchez A, Benfante NE, Reznik E, et al. Urethral Melanoma - Clinical, Pathological and Molecular Characteristics. Bladder Cancer. 2022;8(3):291-301.
- Yelamos O, Merkel EA, Sholl LM, Zhang B, Amin SM, Lee CY, et al. Nonoverlapping Clinical and Mutational Patterns in Melanomas from the Female Genital Tract and Atypical Genital Nevi. J Invest Dermatol. 2016;136(9):1858-65.
- Nassar KW, Tan AC. The mutational landscape of mucosal melanoma. Semin Cancer Biol. 2020;61:139-48.
- Gutierrez-Castaneda LD, Gamboa M, Nova JA, Pulido L, Tovar-Parra JD. Mutations in the BRAF, NRAS, and C-KIT Genes of Patients Diagnosed with Melanoma in Colombia Population. Biomed Res Int. 2020;2020:2046947.
- Castellani G, Buccarelli M, Arasi MB, Rossi S, Pisanu ME, Bellenghi M, et al. BRAF Mutations in Melanoma: Biological Aspects, Therapeutic Implications, and Circulating Biomarkers. Cancers (Basel). 2023;15(16).
- Altieri L, Eguchi M, Peng DH, Cockburn M. Predictors of mucosal melanoma survival in a population-based setting. J Am Acad Dermatol. 2019;81(1):136-42 e2.
- Clavero-Rovira L, Gomez-Tomas A, Bassas-Freixas P, Bodet D, Ferrer B, Hernandez-Losa J, et al. Mucosal Melanoma Clinical Management and Prognostic Implications: A Retrospective Cohort Study. Cancers (Basel). 2024;16(1).
- Jeong YJ, Thompson JF, Ch'ng S. Epidemiology, staging and management of mucosal melanoma of the head and neck: a narrative review. Chin Clin Oncol. 2023;12(3):28.
- Lamichhane NS, An J, Liu Q, Zhang W. Primary malignant mucosal melanoma of the upper lip: a case report and review of the literature. BMC Res Notes. 2015;8:499.
- Cazzato G, Mangialardi K, Falcicchio G, Colagrande A, Ingravallo G, Arezzo F, et al. Preferentially Expressed Antigen in Melanoma (PRAME) and Human Malignant Melanoma: A Retrospective Study. Genes (Basel). 2022;13(3).
- Cascardi E, Cazzato G, Ingravallo G, Dellino M, Lupo C, Casatta N, et al. PReferentially Expressed Antigen in MElanoma (PRAME): preliminary communication on a translational tool able to early detect Oral Malignant Melanoma (OMM). J Cancer. 2023;14(4):628-33.
- Ricci C, Altavilla MV, de Biase D, Corti B, Pasquini E, Molteni G, et al. Unveiling the molecular landscape and clinically relevant molecular heterogeneity of mucosal melanoma of the head and neck region. Histopathology. 2025;87(2):270-83.
- Mokos M, Prkacin I, Gacina K, Brkic A, Pondeljak N, Situm M. Therapeutic Opportunities in Melanoma Through PRAME Expression. Biomedicines. 2025;13(8).
- Scheurleer WFJ, Braunius WW, Tijink BM, Suijkerbuijk KPM, Dierselhuis MP, Meijers RWJ, et al. PRAME Staining in Sinonasal Mucosal Melanoma: A Single-Center Experience. Head Neck Pathol. 2023;17(2):401-8.
- Rambow F, Marine JC, Goding CR. Melanoma plasticity and phenotypic diversity: therapeutic barriers and opportunities. Genes Dev. 2019;33(19-20):1295-318.
- Beigi YZ, Lanjanian H, Fayazi R, Salimi M, Hoseyni BHM, Noroozizadeh MH, et al. Heterogeneity and molecular landscape of melanoma: implications for targeted therapy. Mol Biomed. 2024;5(1):17.
- Curioni-Fontecedro A, Pitocco R, Schoenewolf NL, Holzmann D, Soldini D, Dummer R, et al. Intratumoral Heterogeneity of MAGE-C1/CT7 and MAGE-C2/CT10 Expression in Mucosal Melanoma. Biomed Res Int. 2015;2015:432479.
- Tyrrell H, Payne M. Combatting mucosal melanoma: recent advances and future perspectives. Melanoma Manag. 2018;5(3):MMT11.